# Neutrino at Different Epochs of the Friedmann Universe

Alexandre V. Ivanchik *, Oleg A. Kurichin and Vlad Yu. Yurchenko

Ioffe Institute, Polytekhnicheskaya 26, Saint-Petersburg 194021, Russia; o.chinkuir@gmail.com (O.A.K.); yurchvlad@gmail.com (V.Y.Y.)
* Correspondence: iav@astro.ioffe.ru

**Abstract:** At least two relics of the Big Bang have survived: the cosmological microwave background (CMB) and the cosmological neutrino background (C$\nu$B). Being the second most abundant particle in the universe, the neutrino has a significant impact on its evolution from the Big Bang to the present day. Neutrinos affect the following cosmological processes: the expansion rate of the universe, its chemical and isotopic composition, the CMB anisotropy and the formation of the large-scale structure of the universe. Another relic neutrino background is theoretically predicted, it consists of non-equilibrium antineutrinos of Primordial Nucleosynthesis arising as a result of the decay of neutrons and tritium nuclei. Such antineutrinos are an indicator of the baryon asymmetry of the universe. In addition to experimentally detectable active neutrinos, the existence of sterile neutrinos is theoretically predicted to generate neutrino masses and explain their oscillations. Sterile neutrinos can also solve such cosmological problems as the baryonic asymmetry of the universe and the nature of dark matter. The recent results of several independent experiments point to the possibility of the existence of a light sterile neutrino. However, the existence of such a neutrino is inconsistent with the predictions of the Standard Cosmological Model. The inclusion of a non-zero lepton asymmetry of the universe and/or increasing the energy density of active neutrinos can eliminate these contradictions and reconcile the possible existence of sterile neutrinos with Primordial Nucleosynthesis, the CMB anisotropy, and also reduce the $H_0$-tension. In this brief review, we discuss the influence of the physical properties of active and sterile neutrinos on the evolution of the universe from the Big Bang to the present day.

**Keywords:** cosmology; neutrino; sterile neutrino; lepton asymmetry; baryon asymmetry; primordial nucleosynthesis; CMB; $H_0$-tension





## 1. Introduction

Modern conceptions of the structure of matter are based on the so-called Standard Model of elementary particle physics and fundamental interactions (see e.g., [1]), the confirmation of which was successfully completed with the discovery of the Higgs boson. However, despite the great predictive power of the Standard Model and its numerous experimental confirmations, there are a number of problems that cannot be solved within its framework. A significant portion of these problems are related to observational cosmology. These include the problem of the baryon asymmetry of the universe [2,3], the unknown nature of dark matter [4,5] and dark energy [6,7]. Separately, there is the phenomenon of neutrino oscillations [8,9]: the process of spontaneous transformation of one flavor of neutrino into another, due to the presence of a non-zero neutrino mass. This last circumstance may be the most obvious indication of the need to go beyond the Standard Model.

Modern ideas about the evolution of the universe are based on another standard model: the ΛCDM cosmological model [10,11]. At the same time, the model is adapted from time to time as observational data accumulates and theoretical ideas about the structure of the world develop. Both of these models are closely interconnected today. For example, elementary particles of the SM (and their properties) played a significant, and sometimes decisive role at different stages of the evolution of the universe (see Figure 1).

Having started from a singular state and passed through the inflation stage, in the first moments of the Big Bang our universe entered the radiation-dominated stage of its evolution. Most of the time during this stage, it was neutrinos and photons that determined the dynamics of the expansion of the universe (see Figure 1). This affected the process of Primordial Nucleosynthesis which took place in the first minutes after the Big Bang, during which, in addition to the already existing relic neutrinos (CνB), nonequilibrium antineutrinos appeared from the decays of neutrons and tritium nuclei [12,13]. The discovery of these neutrinos would open a "window" into the first minutes of the hot universe, and would also allow us to test the existence of the baryon asymmetry of the universe on the largest scales.

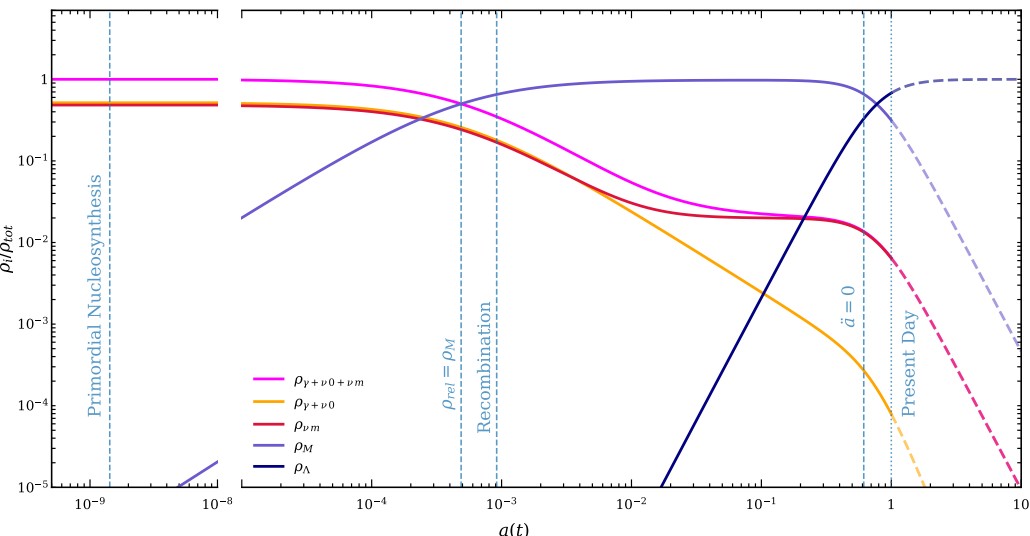

**Figure 1.** The contributions to the total energy density of the universe from different components as a function of the scale factor $a(t)$. The components include photons ($\rho_\gamma$), non-relativistic matter ($\rho_M$), which consists of cold dark and baryonic matter, $\rho_M = \rho_{CDM} + \rho_b$, neutrinos ($\rho_\nu$) and dark energy ($\rho_\Lambda$). The values of the cosmological parameters were taken from the Planck results [14]. Vertical dashed lines mark key cosmological milestones: Primordial Nucleosynthesis, radiation-matter equality ($\rho_{rel} = \rho_M$), Primordial Recombination, and the moment of transition from the decelerating to accelerating expansion of the universe ($\ddot{a}(t) = 0$). For calculations, we utilised the neutrino energy density in the following way: two neutrino flavors are precisely massless ($\rho_{\nu 0}$, yellow curve), and ($\rho_{\nu m}$, red curve), one neutrino flavor has mass $m_\nu = 0.06$ eV (the same as utilized in the Planck analysis [14]). The massless neutrinos behave completely like radiation during the whole course of the evolution of the universe, while the massive one behaves like radiation in the early universe and like non-relativistic matter at later stages.

The next process in which we can see the influence of neutrinos is the process of Primordial Recombination, which took place about 380 thousand years after the Big Bang. At the end of this process, the CMB anisotropy is formed, the observation and analysis of which allows us to obtain high precision estimates of key cosmological parameters [14].

Later, having become non-relativistic, the neutrino increases the contribution to the energy density of the universe of the non-relativistic components, which previously consisted of cold dark matter ($\Omega_{CDM}$) and baryonic matter ($\Omega_b$), and the character of the equation of state changes from relativistic to non-relativistic, so the neutrino component, in a special way, influences the formation of the large-scale structure of the universe.

Possible extensions of the Standard Model of elementary particles suggest the existence of sterile neutrinos. Apparently, the first to introduce the concept of "sterile neutrinos" was Bruno Pontecorvo in 1967 [15]. Their introduction potentially makes it possible to solve not only the problems of generating the masses of the active neutrinos and their oscillations, but also such cosmological problems as the baryon asymmetry of the universe (BAU) and

the nature of dark matter (see, e.g., [16,17]). At the same time, the physical properties of both active and possible sterile neutrinos significantly affect the values of the determined cosmological parameters [18].

The recent results of a number of independent experiments [19,20] indicate the possibility of the existence of a light sterile neutrino ($m_{vs} \sim 1$–3 eV). The presence of such a neutrino is in poor agreement with the predictions of the Standard Cosmological Model, but these contradictions can be removed, for example, by introducing a non-zero lepton asymmetry of the universe, $\xi_v \sim 10^{-2}$, and/or increasing the energy density of active neutrinos. These changes make it possible to reconcile the possible existence of a light sterile neutrino with Primordial Nucleosynthesis, the CMB anistropy, and also reduce the $H_0$-tension[1].

It should be noted that the results of experiments on the detection of light sterile neutrinos are not always consistent with each other. For example, in a recent STEREO collaboration paper [23] the authors reject the hypothesis of the existence of a sterile neutrino on this mass scale. Therefore, the question of the existence of a light sterile neutrino cannot be regarded as finally settled.

Detailed discussions on the influence of neutrino properties on cosmological evolution can be found in large reviews (see, e.g., [24,25]). In our brief review, we emphasise the aspects related to the modern data on active and possible light sterile neutrinos and their influence on various cosmological processes.

Nowadays, there is little doubt about the existence of cosmological neutrinos. The most promising method for their detection is the use of the inverse beta decay of tritium, proposed by S. Weinberg in 1962 [26]. Unfortunately, due to the drastic smallness of the their interaction cross sections at low energies, it has not been possible to register them directly so far. If in the future this can be done, we will directly obtain information about the first seconds, minutes and hours of the early universe.

## 2. The Enigmatic Neutrino

Each of the particles of the Standard Model (see Figure 2) deserves a separate story, but perhaps the most enigmatic particle is still the neutrino, because the explanation of its amazing properties may require going beyond the Standard Model and will have an impact on another Standard Model: the $\Lambda$CDM cosmological model.

Neutrinos do not have an electrical charge; they are born and participate only in weak interactions. There are three generations (flavors) of neutrinos, $v_e, v_\mu, v_\tau$, corresponding to the three generations of charged leptons, electron $e$, muon $\mu$ and tau lepton $\tau$ (Figure 2). In the Standard Model, neutrinos are precisely massless particles, but the phenomenon of neutrino oscillations—the process of spontaneous transformation of neutrinos of one flavor into another—is a direct indication that neutrinos have mass. The explanation of this phenomenon requires an extension of the Standard Model of elementary particles. Although the observed oscillations of neutrinos unambiguously indicate the existence of their mass, it is still not possible to measure it by direct methods, and only lower and upper limits on the sum of the neutrino masses have been experimentally obtained [1]:

$$0.06 \text{ eV} \lesssim \sum m_v \lesssim 0.12 \text{ eV} \qquad (1)$$

The lower limit on the sum of the masses is calculated on the basis of experimental data on the neutrino squared mass differences $\Delta m_{ij}^2 = m_i^2 - m_j^2$, obtained from a number of independent experiments [1]. The upper limit is estimated by analysing the cosmological data [14]. However, the unique properties of neutrinos do not end there; in addition, the following can be pointed out:

- Neutrinos are the second most abundant particles in the universe (after photons). The density of relic photons in the present era is $n_\gamma = 412 \text{ cm}^{-3}$, whereas the density of relic neutrinos (taking into account three flavors of neutrinos and antineutrinos) is $n_v = 336 \text{ cm}^{-3}$.

- It is the lightest known particle with non-zero mass; the neutrino is more than a million times lighter than the electron (see Equation (1)).
- They explicitly break the symmetry of right and left; neutrinos are solely left-handed, antineutrinos are solely right-handed.
- Neutrinos are one of the components of dark matter. Their contribution to the total energy density of dark matter may be up to 1% in the present cosmological epoch.
- Neutrinos have one of the smallest cross sections for interaction with matter ($\sigma \sim 10^{-44}$ cm$^2$ at MeV energies) which determines their enormous penetrating ability, allowing us to see the interiors of stars. In the future, they may allow us to study the first seconds, minutes, and hours of the birth of our universe; the early universe is opaque to electromagnetic radiation.

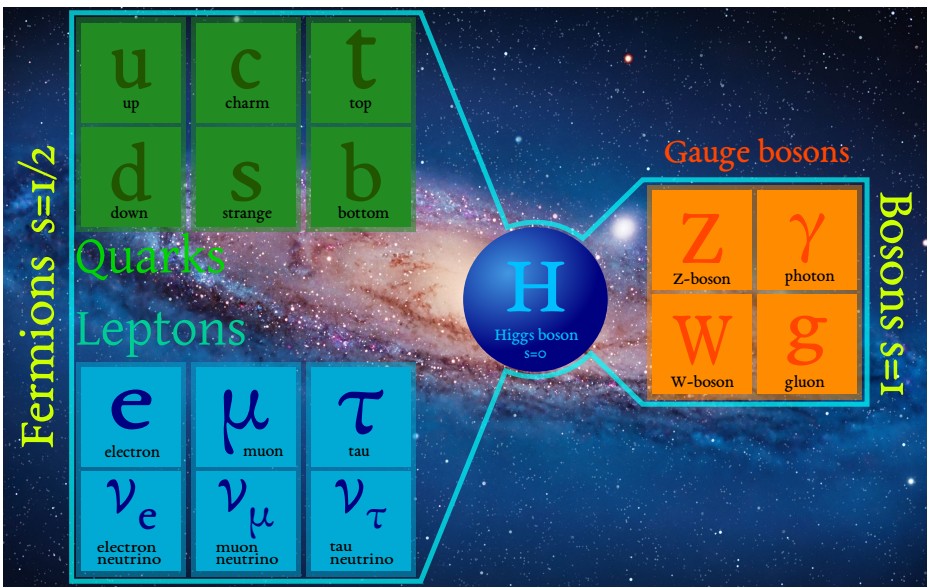

**Figure 2.** The Standard Model of elementary particle physics is the quark-lepton structure of matter. It consists of three generations of fermions. Each generation includes two quarks and two leptons (charged and neutral), which interact with each other via gauge bosons. In this scheme of the quark-lepton structure of matter, phenomena such as the existence of neutrino masses and their oscillations, and the nature of dark matter and dark energy find no explanation.

## 3. Grand Unified Neutrino Spectrum

The first known detection of antineutrinos from nuclear reactors occurred in 1956, about which Fredericks Reines and Clyde Cowan sent a radiogram to Wolfgang Pauli from New York to Zurich. Subsequently, solar neutrinos, atmospheric neutrinos resulting from the interaction of cosmic rays with atmospheric matter (mostly nitrogen and oxygen nuclei), and, finally, ultra-high energy neutrinos ($E_\nu \gtrsim 10^{14}$ eV), which might originate from active galactic nuclei, were recorded.

The energy range over which neutrinos are now observed is indeed enormous: from MeV ($10^6$ eV) to PeV ($10^{15}$ eV) energies (for example, IceCube has detected a few neutrinos with energies above $10^{15}$ eV [27]), but it can be extended by even more than ten orders of magnitude due to the theoretically predicted low-energy cosmological neutrinos[2] ($10^{-4} \lesssim E_\nu \lesssim 10$ eV) and ultrahigh-energy cosmogenic neutrinos ($E_\nu \gtrsim 10^{16}$ eV), the latter arise from the interaction of cosmic rays with the CMB photons, interstellar and intergalactic matter (see, e.g., [28]).

Figure 3 shows the so-called "Grand Unified Neutrino Spectrum"[3], it presents the theoretical and observational spectra of neutrinos of various natures (data used for plotting this spectrum can be found in the paper [13]). It can be seen that the most numerous neutrinos are cosmological ones, which were born in the first moments after the Big Bang. For example, the flux of solar neutrinos at the Earth's surface is 64 billion particles per

square centimetre per second ($6.4 \times 10^{10}$ cm$^{-2}$s$^{-1}$), while the flux of the C$\nu$B neutrinos is at least three trillion particles cm$^{-2}$ s$^{-1}$ ($\gtrsim 3 \times 10^{12}$ cm$^{-2}$s$^{-1}$).

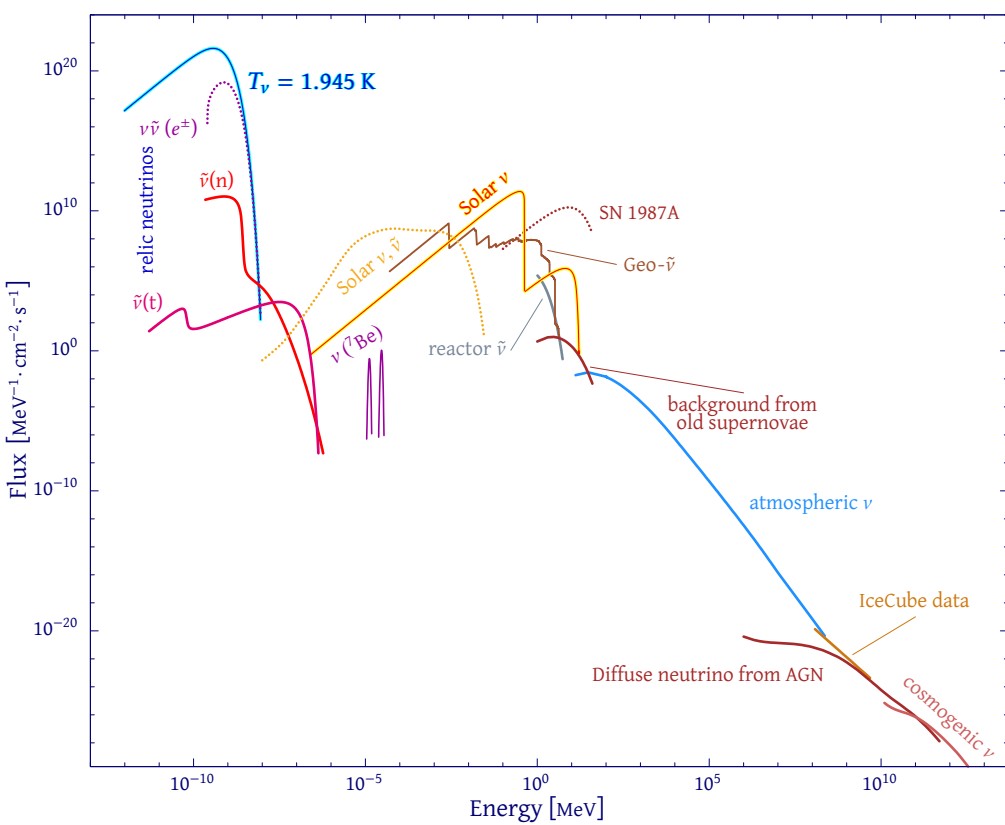

**Figure 3.** Observed and theoretically calculated spectra of neutrinos and antineutrinos generated by various natural phenomena (local and cosmological). For a detailed discussion of all components of the overall spectrum, see [29].

The cosmological neutrinos, like the CMB photons, have a thermal equilibrium spectrum (shown in Figure 4) which for neutrinos is given by the Fermi–Dirac distribution:

$$n_\nu dp = \frac{1}{(2\pi\hbar)^3} \frac{4\pi p^2 dp}{\exp(pc/kT) + 1}. \tag{2}$$

In this paper we will focus on cosmological neutrinos and their impact on various stages of the evolution of the universe.

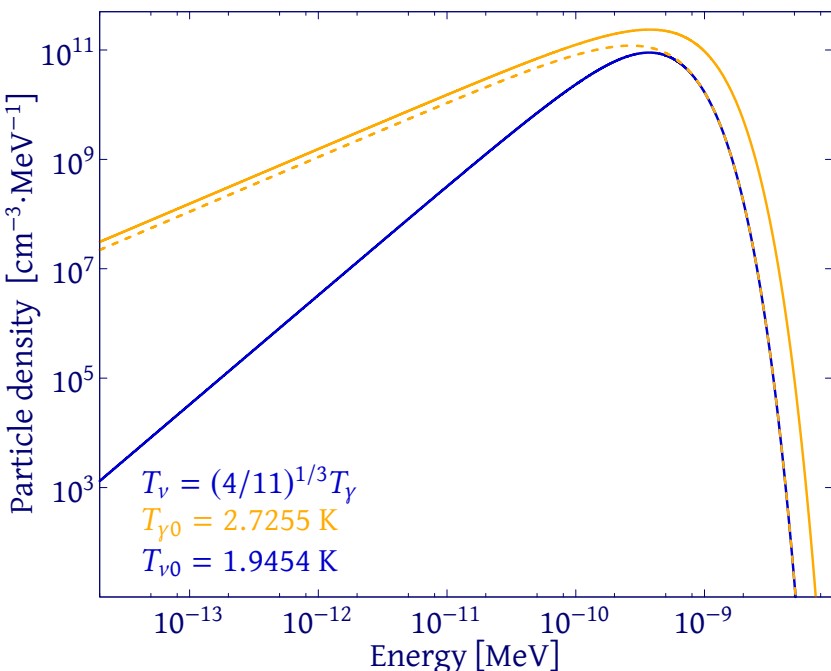

**Figure 4.** The present-day Planckian spectrum of the CMB photons with temperature $T_{\gamma 0} = 2.7255\,\text{K}$ and the Fermi–Dirac spectrum of the C$\nu$B neutrinos with temperature $T_{\nu 0} = 1.9454\,\text{K}$, which are related as follows $T_\nu = (4/11)^{1/3}T_\gamma$ (see e.g., [10]). The photon spectrum is hotter due to electron–positron annihilation that occurred within the first hundred seconds after the Big Bang. The dotted curve shows what the photon spectrum would be with the temperature of the neutrino ($T_{\gamma 0} = T_{\nu 0} = 1.9454\,\text{K}$).

## 4. Cosmological Manifestations of Neutrinos

### 4.1. Radiation-Dominated Epoch, Primordial Nucleosynthesis

In the first moments of the Big Bang, the universe enters the radiation-dominated stage of its evolution, which lasts about 50 thousand years. At this time, neutrinos, along with photons, play a crucial role in the dynamics of the expansion of the universe. Primordial Nucleosynthesis, which took place in the first minutes after the Big Bang, is the earliest moment in the history of the universe that we can probe. As a result of this process, the first lightest nuclei and their isotopes (D, He, Li) appeared, forming the primordial chemical composition of the baryonic matter of the universe. Astronomical observations of the relative abundance of these elements and their comparison with theoretical predictions allow us to estimate one of the key cosmological parameters: the baryon/photon ratio, $\eta = n_b/n_\gamma$. This quantity is related to the baryon density in the universe $\rho_b(\Omega_b)$, as $\eta = 2.74 \times 10^{-8}\,\Omega_b h^2$ (see e.g., [30]). Here $\Omega_b \equiv \rho_b/\rho_c$ is the relative baryon density, $\rho_c = 3H_0^2/8\pi G_N$ is the present critical density, $H_0$ is the present value of the Hubble parameter, $G_N$ is the Newton constant, and $h$, the present value of the Hubble parameter measured in units of 100 km s$^{-1}$ Mpc$^{-1}$.

The most precise estimates of the primordial abundances up to date are as follows:

- Abundance of the primordial $^4$He ($Y_p$) is estimated via the analysis of the spectroscopic samples of dwarf metal-poor galaxies. The analysis yields the estimate $Y_p = 0.247 \pm 0.002$ [31,32].
- Abundance of the primordial D is estimated via the analysis of the quasar spectra containing absorption lines of damped Lyman-alpha (DLA) systems associated with metal-poor intergalactic medium, whose chemical composition is close to the primordial one. The analysis yields D/H $= (2.533 \pm 0.024) \times 10^{-5}$ (see [33] and references therein). There are circumstances that make it difficult to obtain estimates (and their uncertainties) for the abundance of primordial deuterium; a discussion of this problem is presented in [34].

- Abundance of the primordial $^7$Li is estimated via the spectral analysis of metal-poor old stars in the halo of our galaxy. The analysis yields the estimate $^7$Li/H $=$ $(1.6 \pm 0.3) \times 10^{-10}$ [35,36].

Figure 5 shows the calculated abundances of primordial $^4$He, D, $^7$Li as a function of the abundance of baryons in the universe (dark blue lines) and the observed values of the primordial abundances, marked via coloured rectangles.

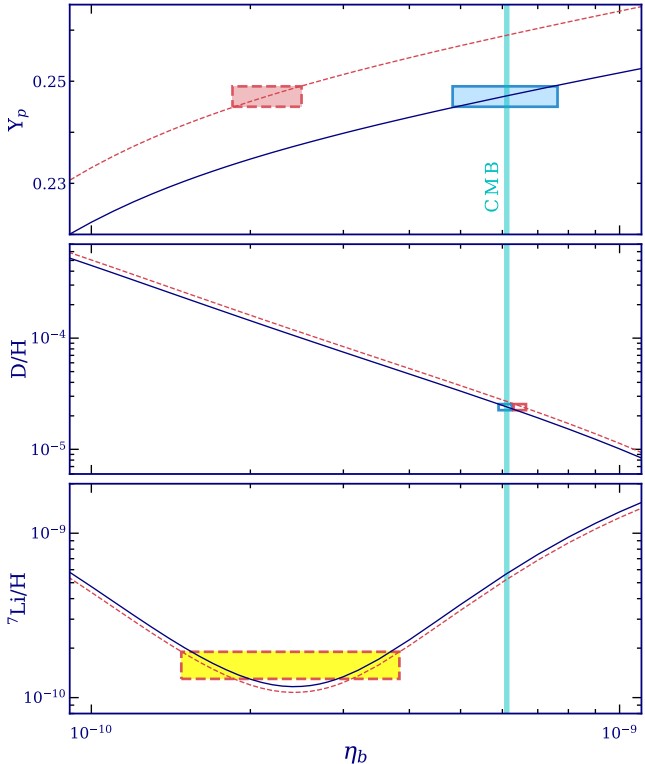

**Figure 5.** Dependence of the primordial abundances $^4$He ($Y_p$), D, $^7$Li on the baryon–photon ratio $\eta$. The dark blue solid lines correspond to the calculated values in the framework of the standard theory of Primordial Nucleosynthesis with three types of active neutrinos ($\Delta N_{\text{eff}} = 0$). Coloured rectangles indicate observed values of primordial abundances. The vertical turquoise line corresponds to the value of $\eta$ estimated as a result of analysis of the CMB anisotropy measured by the Planck satellite [14]. It can be seen that for $^4$He and D the observational data is consistent with the prediction from the CMB anisotropy, while the observed abundance of $^7$Li is significantly lower than the predicted value, which is referred as the "Lithium problem". Red dotted lines and red rectangles correspond to the theory of Primordial Nucleosynthesis in the presence of an additional type of neutrino ($\Delta N_{\text{eff}} = 1$).

Primordial Nucleosynthesis was historically the first way to estimate the total baryon density of the universe, other methods allowed astrophysicists to estimate the number of baryons only in particular astrophysical objects (stars, interstellar and intergalactic gas, galaxy clusters). However, later another independent way to estimate the total baryon density of the universe appeared. It is the analysis of the CMB anisotropy formed during the process of Primordial Recombination, which occurred 380 thousand years after the Big Bang. The primordial abundance estimates based on this method are also shown in Figure 5 with the vertical light blue line. It can be seen that for $^4$He and D the observational data are consistent with the prediction from the CMB anisotropy [14], while the observed value $^7$Li/H $= (1.6 \pm 0.3) \times 10^{-10}$ [35,36] is significantly lower than the predicted value $^7$Li/H $= (4.7 \pm 0.7) \times 10^{-10}$ [37]. The latter is referred to as the "Lithium problem", which still has no explanation.

Independent estimates of the $\eta = n_b / n_\gamma$ based on Primordial Nucleosynthesis and the CMB anisotropy refer to different cosmological epochs. Therefore they make it possible not

only to test the ΛCDM model for self-consistency, but also, in the case of a detected discrepancy, can serve as a tool to search for "physics beyond", which represents a generalisation and extension of the Standard Models of cosmology and particle physics.

### 4.2. Antineutrinos of Primordial Nucleosynthesis

In addition to the relic neutrinos from the Big Bang, antineutrinos of Primordial Nucleosynthesis have recently been theoretically predicted [12,13].

The initial building material for all nuclei synthesised in Primordial Nucleosynthesis are protons and neutrons. The neutrons, in addition to participation in nuclear transformations during collisions with other nuclei, are also subject to spontaneous $\beta^-$-decay ($n \rightarrow p + e^- + \tilde{\nu}_e$). The lifetime of a neutron relative to this process is $\tau_n \simeq 880.2$ s [38]. The electron and antineutrino created in the decay carry away almost all the available decay energy $Q_n \simeq 782.3$ keV [39]. Most decays of neutrons occur after neutrino decoupling (which took place approximately 0.1 s after the Big Bang at temperature $T \sim 2$ MeV), so the antineutrinos produced in these decays are no longer thermalized. Thus, neutron decays during the course of Primordial Nucleosynthesis are a source of non-thermal antineutrinos which will uniformly and isotropically fill the universe at the end of Primordial Nucleosynthesis.

Among the nuclei with noticeable mass fractions that are created in Primordial Nucleosynthesis, there is a nucleus that, like the neutron, is unstable with respect to $\beta^-$ decay. This is the tritium nucleus (T). The lifetime of this nucleus is $\tau_T \simeq 17.66$ years [40], the decay energy is $Q_T \simeq 18.59$ keV [39].

The calculated spectra of antineutrinos from decays of neutrons and tritons in the early universe are presented in Figure 6. The figure shows the spectra of neutrinos and antineutrinos from all sources generating the largest fluxes in the chosen energy range. It can be seen that the antineutrino fluxes of Primordial Nucleosynthesis in the energy range $(10^{-2}$–$10^{-1})$ eV exceed the fluxes from all other sources of neutrinos and antineutrinos. If these nonequilibrium antineutrinos were discovered, we would be able to directly probe the universe in its first minutes and hours after the Big Bang. At the moment, only the CMB studies provide such an opportunity, but this corresponds to a much later cosmological epoch ($\sim$400 thousand years after the Big Bang).

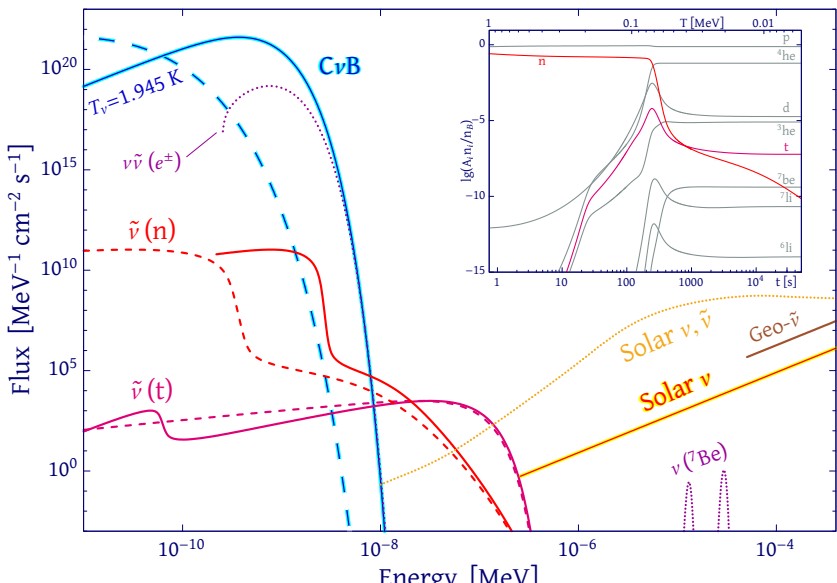

**Figure 6.** Spectra of antineutrinos from $\beta^-$-decays of neutrons (*n*) and tritium nuclei (*t*) (red and red curves). Solid curves show spectra calculated for massless antineutrinos. The dashed curves show the spectra calculated for antineutrinos with mass $m_\nu = 0.01$ eV. Note that there is an energy range where the antineutrino fluxes of Primordial Nucleosynthesis exceed the neutrino fluxes of the Big Bang (blue curve) and solar neutrinos (yellow solid and dotted curves). The figure is based on our works [12,13].

*4.3. Antineutrinos of Primordial Nucleosynthesis as the Probe of Baryon Asymmetry of the Universe*

Direct observational evidence that allows us to conclude that matter predominates in the observable part of the universe and antimatter is absent include the absence of significant annihilation radiation: in the solar system, in our galaxy, and in galaxy clusters, as well as the composition of cosmic rays. The potential observation of relic antineutrinos would allow us to see the most distant, causally unconnected regions to date. The discovery of relic antineutrinos of Primordial Nucleosynthesis could provide evidence of either the baryon asymmetry of most of the visible universe or detect regions with a predominance of antimatter, since the existence of such regions would lead to the generation of relic neutrinos from the decays of antineutrons and antitritium.

## 5. Sterile Neutrinos as an Extension of the Standard Model of Particle Physics and the ΛCDM Cosmological Model

One option for expanding the Standard Model of particle physics is to introduce sterile neutrinos, which do not participate in any Standard Model interactions. The introduction of such particles provides a solution to several problems at once: (i) they make it possible to generate masses of active types of neutrinos (electron, muon and tau neutrinos), (ii) they are suitable for the role of dark matter, (iii) they can become a source for the generation of the baryon asymmetry of the universe (see e.g., [16,17]). At the same time, the mass range of sterile neutrinos is not determined; they can be light, on the order of several eV, and very heavy, up to $10^{15}$ GeV. Their role in relation to cosmology is determined by their mass [41], as follows:

1.  Superheavy sterile neutrinos with masses of $\sim 10^2$–$10^{15}$ GeV. Such neutrinos are capable of generating baryon asymmetry in the early universe through the mechanism described in [42]. Moreover, their lifetime is so short that they will decay even before the Primordial Nucleosynthesis and thermodynamics will "erase" all traces of their existence. Today, only the fact of the presence of baryon asymmetry in the universe could indicate such a possibility (not excluding others).

2.  Heavy sterile neutrinos with masses of $\sim 1$ keV–$10^2$ GeV. Such neutrinos have lifetimes comparable to or longer than the current age of the universe, and are therefore good candidates for cold dark matter particles. In addition, at high temperatures ($\sim 100$ GeV), they can lead to the generation of lepton asymmetry due to oscillations with active neutrinos [43].

3.  Light sterile neutrinos with masses of $\sim 1$ eV–1 keV could have a significant impact on cosmology, which will be discussed in more detail below.

Hints of the possible existence of sterile neutrinos appeared more than ten years ago in various independent experiments (see e.g., [18]), the latest of which talk about the possible existence of light ($m_{\nu,s} \sim 1$–3 eV) sterile neutrinos [19,20]. Such sterile neutrinos may have mixing angles comparable to those of active species. Today, the status of these experiments is quite controversial, since the results obtained do not always agree with each other, and sometimes they are said to be completely inconsistent. For example, in a recent work [23] it is stated that the hypothesis about the existence of a light sterile neutrino could be rejected. However, there is still a region of parameters of oscillations of a light sterile neutrino, which formally does not contradict either the results of the STEREO experiment [23], or the results of the Neutrino-4 experiment [19]. Therefore the question of the existence of a light sterile neutrino cannot be considered finally decided. In addition, sterile neutrinos of electron-volt masses do not fit well into the standard ΛCDM cosmological model. Firstly, the introduction of such neutrinos leads to a discrepancy between the theoretical predictions of Primordial Nucleosynthesis and observational data (see Figure 5 and [37,41]). Secondly, such neutrinos remain relativistic at the epoch of formation of the CMB anisotropy, and, consequently, change the size of the sound horizon, which also leads to a discrepancy between the observational data and the theoretical model (this will be discussed below). Thirdly, such sterile neutrinos are so-called "hot-warm

dark matter" (see, e.g., [11]), and can interfere with cosmological structure formation on small scales, and later become a non-relativistic contribution to it (see e.g., [44]). Fourthly, constraints on the mixing parameters and mass splitting for electronvolt sterile neutrinos, obtained on the basis of cosmological data from Primordial Nucleosynthesis, the CMB anisotropy and baryonic acoustic oscillations (BAO), given in [45], show that in order for the hypothesis of existence of such particles to be consistent with the $\Lambda$CDM model, mixing parameters should be sufficiently small ($|U_{\alpha 4}|^2 \sim 10^{-4}$–$10^{-2}$). This contradicts the experimental results obtained in [19]: $|U_{\alpha 4}|^2 \gtrsim 10^{-1}$. Thus, there is a strong tension between the existence of such sterile neutrinos and the standard $\Lambda$CDM model. However, there is a way out of this situation, and below we provide a discussion of possible options for expanding the $\Lambda$CDM model, which allow us to reconcile cosmological observational data with the hypothesis of the existence of a light sterile neutrino.

*Light Sterile Neutrinos at the Radiation-Dominated Stage and During the Era of Primordial Nucleosynthesis*

The existence of a light sterile neutrino will lead to a change in the energy density at the radiation-dominated stage, which is conveniently parameterized using the so-called effective number of neutrino species $N_{\text{eff}}$, which, in turn, is determined by the relation:

$$\frac{\rho_\nu}{\rho_\gamma} = \frac{7}{8} N_{\text{eff}} \left( \frac{T_\nu}{T_\gamma} \right)^4 = \frac{7}{8} N_{\text{eff}} \left( \frac{4}{11} \right)^{4/3} \tag{3}$$

where $\rho_\nu$ and $\rho_\gamma$ are the energy densities of neutrinos and photons, and the last factor is the fourth power of the ratio of neutrino and photon temperatures, $T_\nu/T_\gamma = (4/11)^{1/3}$ [10]. The contribution to the energy density from three active types of neutrino $N_{\text{eff}} \equiv N_{\text{eff}}^0 = 3.046$ [46,47]. By introducing the value $\Delta N_{\text{eff}}$, which determines the addition to $N_{\text{eff}}$ in the case of the presence of a light sterile neutrino, and taking into account that the temperature of sterile and active neutrinos coincides, we can write the total energy density

$$\rho_R = \rho_\gamma \left( 1 + \frac{7}{8} (N_{\text{eff}} + \Delta N_{\text{eff}}) \left( \frac{4}{11} \right)^{4/3} \right) \tag{4}$$

whence follows:

$$\Delta N_{\text{eff}} = \left( \frac{7}{8} \left( \frac{4}{11} \right)^{4/3} \right)^{-1} \frac{\rho_\nu^{(s)}}{\rho_\gamma} = \left[ \frac{7}{8} \frac{\pi^2}{15} T_\nu^4 \right]^{-1} \frac{1}{\pi^2} \int p^3 f_s(p) dp \tag{5}$$

where $\rho_\nu^{(s)}$ is the energy density of sterile neutrinos, $f_s(p)$ is their distribution function, $T_\nu$ is the temperature of active neutrinos. The form of the distribution function of sterile neutrinos depends on the method of their generation. If sterile neutrinos are produced in some thermal processes, then they would have a Fermi–Dirac distribution function, like active neutrinos. This will lead to $\Delta N_{\text{eff}} = 1$. If the generation of sterile neutrinos is non-thermal (e.g., via the Dodelson–Widrow mechanism [48]), the distribution function may be different. However, calculations carried out in the studies [41,49] using current data on neutrino mixing parameters showed that light sterile neutrinos with a mass of the order of several eVs become completely thermalized by the neutrino decoupling time due to oscillations with the active species. Thus, light sterile neutrinos remain in the expanding universe as an additional relativistic degree of freedom with $\Delta N_{\text{eff}} = 1$.

However, it should be noted that the oscillation parameters of light sterile neutrinos obtained in experiments have quite large uncertainties and thus the reliability of these results requires further confirmation. For example, explicit PMNS matrix for classical 3-flavor and extended 3 + 1-flavor mixing presented in [50] (Section 11) exhibits orders of magnitude larger uncertainties in the matrix elements for 3 + 1 mixing. In the case of significantly smaller mixing angles, the thermalization of sterile neutrinos will be incom-

plete and, therefore, the number of relativistic degrees of freedom will be in the interval $0 \leq \Delta N_{\text{eff}} < 1$ (this case is considered in [18]).

In the case of complete thermalization, $\Delta N_{\text{eff}} = 1$, which significantly changes the expansion rate of the universe and, accordingly, the predictions of Primordial Nucleosynthesis. From Figure 5 (top panel) it is clear that the presence of a light sterile neutrino, which increases the effective number of neutrino species to four, leads to a significant change in the theoretical prediction of the abundance of $^4$He, incompatible with observational data on deuterium and the CMB anisotropy. Nowadays, this is the strictest constraint on the possible existence of a light sterile neutrino.

A solution to this problem can be found in the case of the existence of non-zero lepton (neutrino) asymmetry $L_\nu = (n_\nu - n_{\bar{\nu}})/(n_\nu + n_{\bar{\nu}})$. Here $n_{\nu,\bar{\nu}}$ is the number density of neutrinos and antineutrinos and is defined with the following equation:

$$n_{\nu,\bar{\nu}} = \int \frac{4\pi p^2}{(2\pi\hbar)^3} \frac{dp}{exp\left(\frac{E \mp \mu}{kT}\right) + 1} \tag{6}$$

Direct substitution of number densities shows that the lepton asymmetry $L_\nu$ can be expressed in terms of the dimensionless parameter $\xi = \mu/kT$, where $\mu$ is the chemical potential of the neutrino, as follows:

$$L_\nu = -\frac{1}{\Gamma(3)\left(Li_3(-e^\xi) + Li_3(-e^{-\xi})\right)} \left(\frac{\pi^2}{3}\xi + \frac{\xi^3}{3}\right) \tag{7}$$

where $Li_3(x)$ is the polylogarithm function and $\Gamma(x)$ is the gamma function. For small values of $\xi$ this gives:

$$L_\nu \approx \frac{\pi^2}{9\zeta(3)}\xi\left(1 + \frac{\xi^2}{\pi^2}\right) \approx 0.91 \times \xi \tag{8}$$

For temperatures above 2 MeV weak interaction reactions proceeded intensively, including those that determined the neutron–proton ratio:

$$\begin{aligned} n + e^+ &\longleftrightarrow p + \tilde{\nu}_e \\ n + \nu_e &\longleftrightarrow p + e^- \\ n &\longleftrightarrow p + e^- + \tilde{\nu}_e \end{aligned} \tag{9}$$

As long as the rate of weak interactions exceeds the expansion rate of the universe, it enables the neutron–proton ratio to track its equilibrium value:

$$\left(\frac{n}{p}\right)_{eq} = exp\left(-\frac{\Delta m}{kT}\right) \tag{10}$$

where $n$ and $p$ are number densities of neutron and protons, $\Delta m = m_n - m_p = 1.293$ MeV.

For temperatures below 2 MeV the rate of weak interactions becomes less than the rate of expansion of the universe, which leads to the removal of the neutron–proton mixture from thermodynamic equilibrium with the primordial plasma (so-called "neutron freeze-out" [37]). Up to the start of Primordial Nucleosynthesis the neutron–proton ratio decreases slowly due to the $\beta$-decay of free neutrons. The evolution of neutron density can be described via the equation [10]:

$$\frac{dX_n}{dt} = -\lambda(n \to p)X_n + \lambda(p \to n)(1 - X_n) \tag{11}$$

where $X_n = n/(n + p)$. In the equation, the $\lambda$ are the rates of the corresponding reactions, which depend on the lepton asymmetry (explicit formulae can be found in [51]). The presence of non-zero lepton asymmetry leads to an increase in the rate $\lambda(n \to p)$ and a decrease in the rate $\lambda(p \to n)$. These reactions in conjunction lead to an overall decrease

of the neutron–proton ratio, which in turns leads to decreased production of $^4$He during Primordial Nucleosynthesis. In the case of non-zero lepton asymmetry the neutron–proton ratio shifts from its equilibrium value (10) [52]:

$$\frac{n}{p} = exp\left(-\frac{\Delta m + \mu}{kT}\right) = \left(\frac{n}{p}\right)_{eq} \times exp(-\xi) \tag{12}$$

It is worth noting, however, that the presence of non-zero lepton asymmetry itself leads to an increase in the total energy density of ultrarelativistic particles, and, consequently, the expansion rate of the universe. The increase of the expansion rate in turn leads to higher abundance of primordial $^4$He. Direct calculation shows that this effect is quite small compared to the previously discussed one [52], and thus the higher $^4$He yield associated with the increased expansion rate is offset by the lower neutron–proton ratio at the start of Primordial Nuclosynthesis.

Of all the primordial elements, $^4$He is the most sensitive element to the neutron–proton ratio [30]. This is due to the fact that all the neutrons that existed at the start of Primordial Nucleosynthesis will either decay to protons, or form primordial nuclei (mainly $^4$He ones). Thus the total abundance of primordial helium can be approximated as

$$Y_p \approx \frac{2\,n/p}{1 + n/p} \tag{13}$$

where $n/p$ is neutron–proton ratio at start of Primordial Nucleosynthesis, which is evaluated using Equation (11). In the case of non-zero lepton asymmetry this equation transforms to

$$Y_p \approx \frac{2\,(n/p)_{eq} \times e^{-\xi}}{1 + (n/p)_{eq} \times e^{-\xi}} \tag{14}$$

For small values of $\xi$ this equation simplifies:

$$Y_p \approx Y_p^{eq}\left[1 - \xi\left(1 - \frac{Y_p^{eq}}{2}\right)\right] \tag{15}$$

where $Y_p^{eq}$ is the value of $Y_p$ with zero lepton asymmetry and is a function of $\eta$ and the effective number of neutrino species $N_{\text{eff}}$. Thus, the presence of the lepton asymmetry leads to a decreased abundance of primordial $^4$He.

Using the described methodology for taking non-zero lepton asymmetry into account, we calculated the abundance of primordial $^4$He as a function of baryon–photon ratio $\eta$, $N_{eff}$ and $\xi$. The results are presented on the left panel of the Figure 7. In the calculation we considered three cases: standard Primordial Nucleosynthesis with $\Delta N_{eff} = 0$ and $\xi = 0$ (dark blue curve), Primordial Nucleosynthesis with $\Delta N_{eff} = 1$ and $\xi = 0$ (red curve), and Primordial Nucleosynthesis $\Delta N_{eff} = 0$ and $\xi \neq 0$ (yellow curve). Fitting the theoretical calculation to the observed abundance of primordial helium from [32], we found that the increase of the $Y_p$ associated with the presence of a fully-thermalized light sterile neutrino can be completely compensated by the non-zero lepton asymmetry with the value $\xi = 0.052 \pm 0.001$.

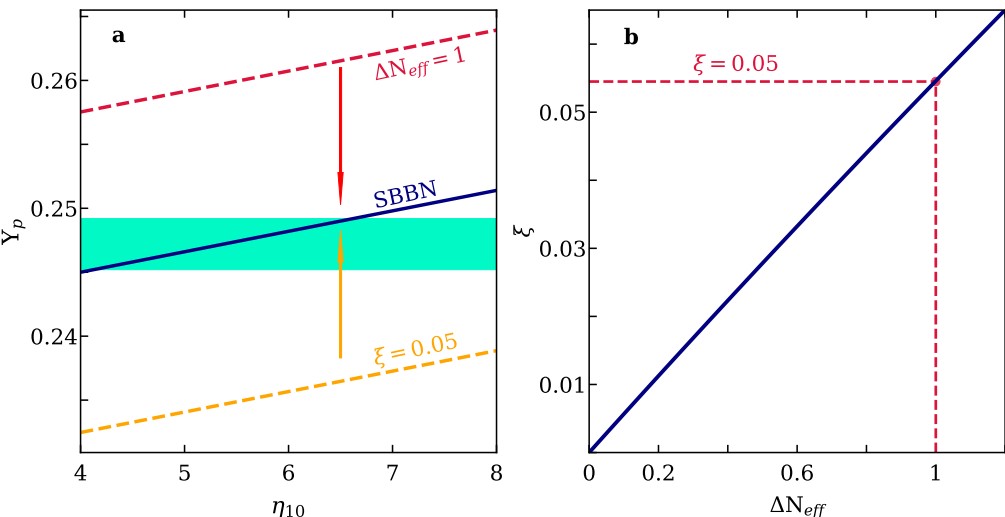

**Figure 7.** (**a**) The calculated dependencies of $Y_p$ on the baryon-to-photon ratio $\eta_{10}$ ($\eta_{10} = \eta \times 10^{10}$). The red dashed curve shows the dependence $Y_p(\eta_{10})$ in the presence of one light sterile neutrino (compare with Figure 5). The yellow dashed curve shows the dependence $Y_p(\eta_{10})$ in the presence of lepton asymmetry $\xi_e = 0.05$. The dark blue curve shows the dependence $Y_p(\eta_{10})$ in the standard case. The cyan stripe indicates the observed abundance of $^4$He taken from [31]. (**b**) The panel shows the value of lepton asymmetry $\xi$, which allows one to completely compensate for the influence of the additional relativistic degree of freedom associated with a light sterile neutrino.

## 6. The Influence of Neutrinos on the Formation of the CMB Anisotropy

After the radiation-dominated era ($\sim$50 thousand years after the Big Bang), comes the era of dominance of non-relativistic matter: cold dark and baryonic matter. Then, after approximately 7 billion years, the universe transits from the decelerating to accelerating expansion, while the neutrino affects the dynamics of the expansion of the universe at each stage of the evolution. The expansion rate of the universe, characterised by the Hubble parameter, is defined via the following equation:

$$\mathrm{H}(a) \equiv \frac{1}{a}\frac{da}{dt} = \mathrm{H}_0 \sqrt{\Omega_\Lambda + \Omega_{\mathrm{cdm}}a^{-3} + \Omega_{\mathrm{b}}a^{-3} + \Omega_\gamma a^{-4} + \sum_\nu \Omega_\nu f_\nu(a)} \qquad (16)$$

where $H_0$ is the current value of the Hubble parameter, $\Omega_{\mathrm{CDM}}, \Omega_{\mathrm{b}}, \Omega_\gamma, \Omega_\nu$ and $\Omega_\Lambda$ are the fractions of energy densities of cold dark matter, baryons, photons, neutrinos and dark energy in the universe at the moment, and the functions $f_\nu(a)$ determine the dependence of the neutrino contribution on the scale factor of the universe, i.e., in the corresponding cosmological era.

Figure 8, taken from our paper [18], presents the effective number of neutrino species, taking into account the possible existence of a light sterile neutrino. It can be seen that in the early stages of the evolution of the universe, all neutrinos are relativistic and therefore make a significant contribution to the energy density and the expansion rate of the universe, which in turn determines the size of the sound horizon at the time of Primordial Recombination. Sterile neutrinos with a mass of 2.7 eV [19] become non-relativistic before the recombination and even earlier at the radiation-dominated stage, therefore it can be classified as "warm" dark matter. Sterile neutrinos with a mass of 1 eV [20] become non-relativistic during the transition from the radiation-dominated to the matter-dominated stage, therefore it is "hot-warm" dark matter. Active neutrinos with masses less than 0.1 eV become non-relativistic after the recombination, therefore they compose "hot" dark matter. All these factors influence the formation of the CMB anisotropy, the study of which makes it possible to obtain estimates of cosmological parameters with unprecedented accuracy.

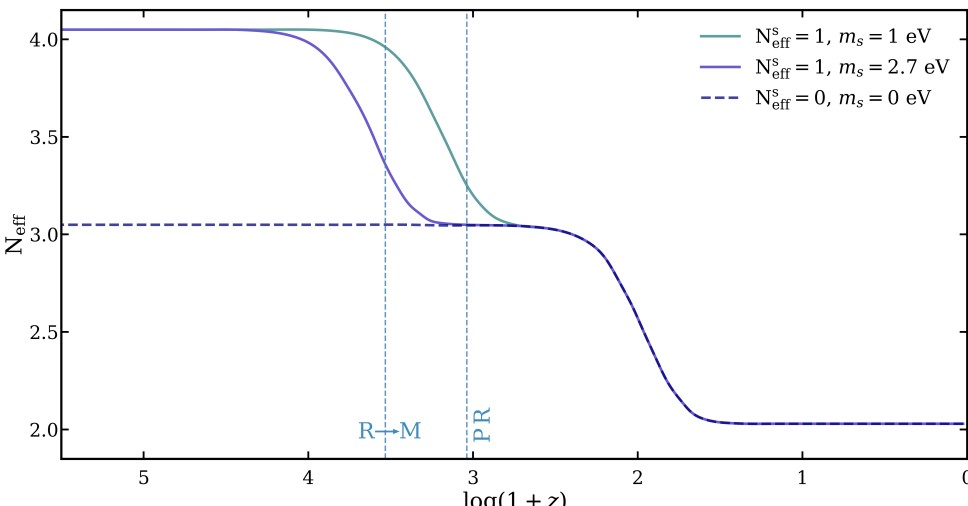

**Figure 8.** The effective number of relativistic neutrino species taking into account the possible existence of a light sterile neutrino as a function of cosmological redshift $z$. Vertical dashed lines represent the moment of transition from the radiation to the matter dominated stage (R→M) and the moment of Primordial Recombination (P R). It can be seen that in the early stages of the evolution of the universe, all neutrinos are relativistic. Sterile neutrinos with a mass of 2.7 eV [19] become non-relativistic before the recombination and even earlier at the radiation-dominated stage, therefore it can be classified as "warm" dark matter. Sterile neutrinos with a mass of 1 eV [20] become non-relativistic during the transition from the radiation-dominated to the matter-dominated stage, therefore it is "hot-warm" dark matter. Active neutrinos with masses less than 0.1 eV become non-relativistic after the recombination, therefore they compose "hot" dark matter.

An analysis of the CMB anisotropy (Planck Collaboration, [14]) within the standard spatially-flat 6-parameter ΛCDM cosmology allows us to estimate key cosmological parameters: $\Omega_b$, $\Omega_{CDM}$, $\theta_*$, $n_s$, $A_s$, $\tau$, which represent the present-day values of the baryon and cold dark matter densities, the angular size of the sound horizon, the scalar spectral index, the amplitude of scalar perturbations, and the optical depth of the reionized plasma, respectively. In turn, using the obtained values of these six parameters it is possible to determine a number of other cosmological quantities (for instance, the Hubble constant $H_0$, the age of the universe $t_0$, the dark energy density $\Omega_\Lambda$).

The possible existence of sterile neutrinos or/and changes in the physical parameters of active neutrinos (for instance, $T_\nu \neq T_{\nu 0}$ or $L_\nu \neq 0$) can be included into an analysis of the CMB anisotropy as additional free parameters. It will lead to a noticeable redistribution of estimates of other cosmological parameters while the model remains consistent with the observational data on the CMB anisotropy [18]. This effect is shown in Figure 9 and in Table 1. To assess this effect, the CMB anisotropy data was analysed in the following cases: standard ΛCDM model, a ΛCDM model with an additional light sterile neutrino, and a ΛCDM model with non-standard active neutrino temperature [18]. For the analysis of the CMB anisotropy the neutrinos were treated in the same way as described in the Planck Collaboration paper [14]: the analysis assumes a normal neutrino mass ordering and active flavors with masses $m_1 = m_2 = 0$ and $m_3 = 0.06$ eV. Additionally, in the case of a 3 + 1 neutrino mixing scenario, a light sterile neutrino with a mass 2.7 eV [19], or with a mass of 1 eV [20] was included. The upper panel of Figure 9 demonstrates the effect of a straightforward addition of sterile neutrinos or a non-standard temperature of active neutrinos into a model in which other cosmological parameters are fixed at standard values. It is easy to see that changing the ΛCDM model in this way leads to a very strong discrepancy between the fit and observational data. It should be noted that similar analyses of the CMB anisotropy have been carried out previously (see e.g., [45,53–55]). In these studies, the authors derived constraints on the parameters of sterile neutrinos based on the CMB anisotropy and other cosmological data. This was achieved either by fixing values of

the fitted parameters or by imposing certain priors on them. Thus the resulting constraints on the sterile neutrinos and the effective number of relativistic degrees of freedom turned out to be relatively small. Lower panels of Figure 9 demonstrate fits of the CMB anisotropy data in the following cases: (b) the standard ΛCDM model, (c) ΛCDM with the inclusion of a light sterile neutrino, and (d) ΛCDM with the modified value of active neutrino temperature. In all three cases, all of the fitted parameters, including six standard ones and additional parameter associated with the corresponding effect, were set free. It can be seen that all three fits are in a good consistency with the observational data on the CMB anisotropy. This consistency is achieved via redistribution of all estimates of the model parameters (see Table 1).

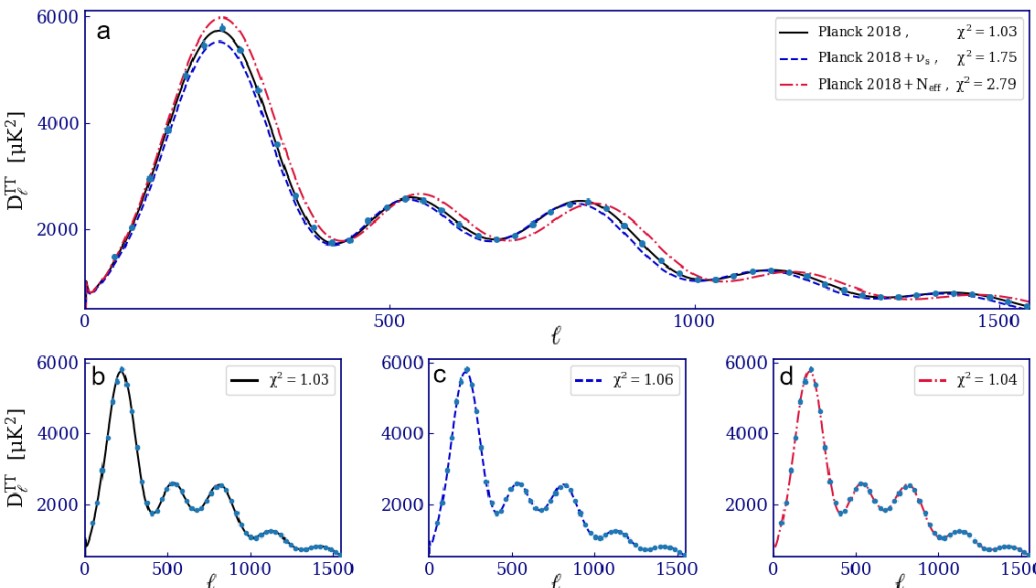

**Figure 9.** Observational data on the CMB anisotropy (blue dots) and the results of its fitting with theoretical models. The goodness-of-fit can be assessed using the reduced $\chi^2$ value, which is one of the output values of the Planck Collaboration code [14]. (**a**) The best-fits to the observational data are for models with all standard cosmological parameters having their values fixed. The solid black curve represents the standard ΛCDM model, the dashed blue curve represents the standard ΛCDM model with the addition of a light sterile neutrino (for a total of four neutrino flavors), the dashed red curve represents the standard ΛCDM model with the additional relativistic degree of freedom, but with three neutrino flavors. (**b**) The black curve represents the best-fit curve for the standard ΛCDM model (the same as on panel (**a**)). (**c**) The dashed blue curve represents the best-fit for the ΛCDM model with the addition of a light sterile neutrino, and all key cosmological parameters set free. (**d**) The dashed red curve represents the best-fit for the ΛCDM model with the additional relativistic degree of freedom, and all cosmological key parameters set free. Thus, the hypothesis of the existence of light sterile neutrinos does not contradict the observed CMB anisotropy.

The resulting estimates of the key cosmological parameters are presented in Table 1. It should be noted, that both of the non-standard cases can be parameterised in terms of the effective number of neutrino species $N_{eff}$. While in the third and in the fourth columns $N_{eff}$ have similar values (4 and 3.9 correspondingly), the physical reasons behind the values are completely different. In the third column $N_{eff}$ equals 4 due to a physical presence of additional particle; a completely thermalised light sterile neutrino. In the fourth column $N_{eff}$ equals 3.9 due to the higher temperature of the three active flavors $T_\nu = 2.07$ K (see Equation (3)). This fact leads to different cosmological consequences. The inclusion of a light sterile neutrino not only noticeably changes the estimates of the cosmological parameters, but also significantly worsens the Hubble tension. On the other hand, the consideration of a non-standard temperature of active neutrinos leads only to a slight redistribution of the cosmological parameters (compare the fourth and the

second columns). Moreover, the increase of the temperature of active neutrinos leads to agreement between the CMB estimate of $H_0 = 72.81 \pm 0.62$ km s$^{-1}$ Mpc$^{-1}$ [18] and the "late" estimate of the $H_0 = 73.04 \pm 1.04$ km s$^{-1}$ Mpc$^{-1}$ [21] (i.e., to the solution of the Hubble tension). In the analyses presented in [18] it was found that to fully agree between the two estimates, the neutrino temperature needs to be $T_\nu = 2.07$ K, which is slightly higher than the standard value of $T_{\nu 0} = 1.94$ K [10]. There are different mechanisms for active neutrino heating, for example it can be due to a decay of keV-mass sterile neutrinos during the whole course of the evolution of the universe [18], a decay of MeV-mass sterile neutrinos before Primordial Nucleosynthesis [56], or other special mechanisms.

Thus, the introduction of an eV-mass sterile neutrino into the $\Lambda$CDM model is consistent with the CMB anisotropy, but at the same time it enhances the Hubble tension. This problem, in turn, may be solved by a heating of the active neutrinos, which in a minimal way can be provided by the decays of heavy sterile neutrinos that compose dark matter.

**Table 1.** Dependence of estimates of the cosmological parameters on the effective number of neutrino species $N_{\text{eff}}$ and the present-day neutrino temperature $T^0_{C\nu B}$. The second column contains estimates obtained for the standard $\Lambda$CDM model in the Planck Collaboration analyses [14]. The third column contains estimates obtained for the $\Lambda$CDM model with three active and one light sterile neutrino mixing. The last column contains estimates obtained for the case of the $\Lambda$CDM model with only three active neutrinos with non-standard temperature. The parameters are evaluated for a light sterile neutrino with mass $m_\nu = 2.7$ eV. The estimates of $\Omega_i$ are given in percentages (as $100\% \times \Omega_i$) and $H_0$ is given in units km s$^{-1}$ Mpc$^{-1}$. The values of $\Omega_{\text{CDM}}$ and $\Omega_\Lambda$ are defined as follows: $\Omega_{\text{m}} = \Omega_{\text{CDM}} + \Omega_{\text{b}} + \Omega_\nu$, $\Omega_\Lambda = 1 - \Omega_{\text{m}}$.

| Parameter | Planck 2018 | $T^0_{C\nu B}$ = 1.94 K $N_{\text{eff}}$ = 4 | $T^0_{C\nu B}$ = 2.07 K $N_{\text{eff}}$ = 3.9 |
|---|---|---|---|
| $\Omega_{\text{CDM}}$ | $26.45 \pm 0.50$ | $32.36 \pm 0.57$ | $24.92 \pm 0.49$ |
| $\Omega_{\text{b}}$ | $4.93 \pm 0.09$ | $5.88 \pm 0.11$ | $4.33 \pm 0.08$ |
| $\Omega_\nu$ | $0.14$ | $7.58$ | $0.15$ |
| $\Omega_{\text{m}}$ | $31.53 \pm 0.73$ | $45.8 \pm 1.1$ | $29.41 \pm 0.87$ |
| $\Omega_\Lambda$ | $68.47 \pm 0.73$ | $54.2 \pm 1.1$ | $70.58 \pm 0.87$ |
| $H_0$ | $67.36 \pm 0.54$ [1] | $62.20 \pm 0.53$ | $72.81 \pm 0.62$ |

[1] The "Late" $H_0$ measurement (independent of the $\Lambda$CDM model) carried out by SH0ES collaboration gives $H_0 = 73.04 \pm 1.04$ km s$^{-1}$ Mpc$^{-1}$ [21]. A significant discrepancy between "late" and CMB measurments of $H_0$ is referred to as $H_0$-tension.

## 7. The Influence of Neutrinos on Subsequent Stages of the Evolution of the Universe

The influence of neutrinos in the late stages of the evolution of the universe is to participate in the formation of the large-scale structure of the universe and is determined by which class of dark matter the neutrino belongs to. Active neutrinos with masses in the range $0.06$ eV $\lesssim \sum m_\nu \lesssim 0.12$ eV remain relativistic at the matter-dominated stage even after the primordial recombination (see Figure 8). Moreover, the light neutrino component (for instance, for the normal neutrino mass hierarchy $m_1 < m_2 \ll m_3$, when $m_1 < m_2 \ll 0.06$ eV) remains relativistic up to the present day. Such neutrinos are a component of "hot" dark matter and their effect is the damping of structure formation on small scales. Heavy sterile neutrinos with masses of the order of keV or more can compose warm and cold dark matter. These cases of light active neutrinos and heavy sterile neutrinos, in terms of their influence on the formation of the large-scale structure of the universe, have been analyzed in detail many times and can be considered as the standard cases (see e.g., [44] and references therein). At first view, the case of a light sterile neutrino ($m_{\nu s} \sim 1$–3 eV) could be referred to the first case of active neutrinos from the point of view of the formation of the large-scale structure, but the reconciliation of the existence of such a neutrino with the CMB anisotropy leads to a significant redistribution of the values of the key cosmological parameters (see Section 6), which in turn should be taken into account when modeling the large-scale structure of the universe.

## 8. Conclusions

Neutrino astronomy has opened up new opportunities for us to study the universe in the diversity of its manifestations. Cosmological relic neutrinos, born in the very first moments of the Big Bang, participate in all stages of the evolution of the universe, making a significant contribution to the dynamics of its expansion, in contrast to photons, whose energy density dominates only in the early stages, or from dark matter and dark energy, whose contribution becomes significant only in later epochs.

The possible existence of a light sterile neutrino ($m_{\nu s} \sim$ 1–3 eV) is in poor agreement with the predictions of the Standard Cosmological Model. It contradicts the predictions of both the Primordial Nucleosynthesis and the CMB anisotropy data. However, these contradictions can be removed by an extension of the Standard Cosmological Model, for example, by introducing a non-zero lepton asymmetry of the universe $\xi_\nu \sim 10^{-2}$ or additional relativistic degrees of freedom. Additionally, the CMB anisotropy data can be reconciled with the possible existence of a light sterile neutrino by changing the values of the key cosmological parameters. We show that it leads to a significant redistribution of the constituent components of matter $\Omega_i = \rho_i / \rho_c$ (within 10%). This fact must be taken into account in the later stages of the evolution of the universe, namely when modelling the formation of the large-scale structure of the universe.

The spectrum of the antineutrinos of Primordial Nucleosynthesis contains additional information about the course of non-equilibrium processes in the early universe, in the first minutes and hours after the Big Bang. Additionally, its detection would allow us to test the baryonic asymmetry of the universe on the largest scales, since the existence of antimatter-dominated regions would lead to the generation of relic neutrinos from the decays of antineutrons and antitritium.

If in the future it will be possible to detect cosmological neutrinos, due to their very high penetrating power, we will directly obtain information about the first seconds, minutes and hours of the evolution of the universe after the Big Bang.

**Author Contributions:** A.V.I. provided main idea, conceptualisation, provided general supervision and project management, and performed overall validation and control. V.Y.Y. and O.A.K. carried out formal analysis, provided software for the calculations required for the paper, and worked on visualization of the results. All authors contributed equally to the initial draft of the paper. The review and forthcoming editing were performed by A.V.I. and O.A.K. All authors have read and agreed to the published version of the manuscript.

**Funding:** This research was funded by RSF grant number 23-12-00166.

**Conflicts of Interest:** The authors declare no conflicts of interest.

## Notes

[1]  The $H_0$-tension is the most statistically significant deviation in modern cosmology ($\sim 5\sigma$ CL), which is the discrepancy between the estimates of the Hubble parameter obtained on cosmological data [14] and "late" model-independent measurements of $H_0$ in the local universe [21]. A detailed discussion of possible solutions can be found in the review [22].

[2]  In reality, cosmological neutrinos are a mixture of neutrinos and antineutrinos, however, in the scientific literature, where this does not lead to confusion, neutrinos and antineutrinos are commonly referenced as neutrinos.

[3]  The title is taken from [29].

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
