# Peer review of "Neutrino at Different Epochs of the Friedmann Universe"

_universe, doi:10.3390/universe10040169_

Round 1
Reviewer 1 Report
Comments and Suggestions for Authors
The paper briefly reviews the state of neutrinos in cosmology, namely their role and phenomenology in the nucleosynthesis, recombination and structure formation processes. In general I find the paper covers most of the effects related to neutrino in cosmology, though too brief in some aspects. Before considering for publication, the following issues need to be addressed:
Major Points:
1. Reference is not sufficient. Many claims in the paper lack sufficient references. For example:
* Line 26, please provide at least one reference (e.g. review) for baryon asymmetry and neutrino oscillation.
* Line 64, please provide reference for the claim that introducing lepton asymmetry might lift the constraint on a light sterile neutrino.
* Line 67, please comment on and provide reference for the Hubble tension.
* Line 124, please comment on and provide reference for the appearance of ultrahigh-energy neutrinos.
* Line 251, please provide reference for the constraint on eV massive neutrino in LCDM.
* Line 279, please quote actual numbers from reference to backup the statement.
The above list is not complete. I suggest the authors to check and add necessary references throughout the paper carefully.
2. Massive neutrinos' effect on structure formation is a very important cosmological phenomenon. It is in fact the main source of constraint on neutrino total mass from cosmology. Also CMB provides stringent constraint on free streaming particles, e.g. number of neutrino species, during recombination. It is one of the main source of cosmological constrain on N_eff. A review of neutrinos in different epochs of the Universe should at least cover these two aspects with substantial text. However, they are only very briefly mentioned in the paper compared with other aspects. If it is the authors' intention to focus on the radiation dominating era and nucleosynthesis process, please modify the title and abstract accordingly to be more specific.
3. In section 5.2 and Fig.8, Tab.1 some results are presented regarding the effect of a massive sterile neutrino on CMB spectra. Please further clarify the cosmological setup. Currently it is unclear to me how they treat neutrinos. Is it 3 massless+1 massive with mass 2.7eV or 2 massless+2 massive with mass 0.06eV and 2.7eV? What does the N_eff in Tab.1 mean? Is it the N_eff when all neutrino species are relativistic? Why different neutrino temperature in Tab.1? What does each panel of Fig.1 refer to and how is the chi^2 in legend calculated? etc.
Minor Points:
1. As far as I know, nucleosynthesis is often used as one word without primordial. Is there a special reason for adding primordial before it?
2. To my knowledge, increasing N_eff to 4 can lift the Hubble constant from CMB in LCDM to a larger value (so smaller H0 tension). However it is stated in line 329 that adding one light sterile neutrino makes the tension worse. Can the author further comment on what is difference here?
3. It seems Tab.1 reports 100*Omega_i rather than Omega_i .
4. In the footnote below Tab.1, SH0ES measurement of H0 is not entirely independent of LCDM. In order to extract H0 from the observed luminosity distance d_L(z), a model of d_L(z) needs to be assumed.
Author Response
We thank the referee for carefully reading our article and for providing important and useful comments and questions.
Our additions and corrections to the paper are marked with red colored text.
Major Points
- Reference is not sufficient. Many claims in the paper lack sufficient references. For example: …..
In accordance with the reviewer's comment, we have added a large number of additional references, explanations and discussions in various parts of the paper where the referee asked for it and where it was necessary.
As for the direct reviewer's comments:
- Line 26, please provide at least one reference (e.g. review) for
baryon asymmetry and neutrino oscillation.
We have provided the corresponding reference
- Line 64, please provide reference for the claim that introducing
lepton asymmetry might lift the constraint on a light sterile neutrino.
The idea that presence of a non-zero lepton asymmetry may lead to a decrease of primordial helium abundance was proposed earlier e.g. by Gary Steigman. We worked further on this idea and found (based on our independent estimate of Yp), that actual value of asymmetry should be ~0.05. This actually was our own result, that we obtained in our calculations. So we have included an extended discussion with all required reference to the subsection 5.1
- Line 67, please comment on and provide reference for the Hubble
We have added a footnote with the comments and reference.
- Line 124, please comment on and provide reference for the
appearance of ultrahigh-energy neutrinos.
We have added a corresponding reference to the Section 3, and added additional Figure 4 which shows spectra of cosmological neutrinos and photons.
- Line 251, please provide reference for the constraint on eV
massive neutrino in LCDM.
We have added and extended explanation of this statement to the Section 5, and provided a reference for constraints on mass splitting a mixing parameters in a 3+1 mixing scenario in LCDM.
- Line 279, please quote actual numbers from reference to backup the
We have added a reference to the paper by A.P. Serebrov where PMNS matrices in standard 3-neutrino and 3+1 neutrino mixing scenarios are presented. We decided not to include these matrices explicitly in our review, since this requires additional detailed explanations on elementary particle physics. In our review, we focus more on the cosmological manifestations of these particles.
- Massive neutrinos' effect on structure formation is a very important cosmological phenomenon. It is in fact the main source of constraint on neutrino total mass from cosmology. Also CMB provides stringent constraint on free streaming particles, e.g. number of neutrino species, during recombination. It is one of the main source of cosmological constrain on N_eff. A review of neutrinos in different epochs of the Universe should at least cover these two aspects with substantial text. However, they are only very briefly mentioned in the paper compared with
other aspects. If it is the authors' intention to focus on the radiation dominating era and nucleosynthesis process, please modify the title and abstract accordingly to be more specific.
We agree with the referee on the topic, but it is important to note these constraints are model-dependent and are obtained in case of the fixed cosmological parameters. We sufficiently extended explanations concerned with the CMB anisotropy and added Section 7 with a discussion of the formation of large scale structure of the Universe.
- In section 5.2 and Fig.8, Tab.1 some results are presented regarding the effect of a massive sterile neutrino on CMB spectra. Please further clarify the cosmological setup. Currently it is unclear to me how they treat neutrinos. Is it 3 massless+1 massive with mass 2.7eV or 2 massless+2 massive with mass 0.06eV and 2.7eV? What does the N_eff in Tab.1 mean? Is it the N_eff when all neutrino species are relativistic? Why different neutrino temperature in Tab.1? What does each panel of Fig.1 refer to and how is the chi^2 in legend calculated? etc.
We have added an extended explanation and clarification of all the reviewer's questions in section 6. There we have also added an explanation related to one of the minor points concerned with solving the Hubble tension by increasing Neff.
Minor Points
- “As far as I know, nucleosynthesis is often used as one word without primordial. Is there a special reason for adding primordial before it?”
We emphasize the use of the term Primordial Nuclesynthesis (also referenced to as Standard Big Bang Nucleosynthesis) in a relation to a specific cosmological epoch when the first chemical elements has been formed. The main point here is that this term does not only mean a set of specific nuclear reactions and related processes, but also a specific time interval, which is characterized by its physical properties (temperature, reaction rate, etc.), which took place in the Early Universe. Thus, we use the term “Primordial Nucleosynthesis” in order to distinguish it from stellar, galactic, etc. nucleosynthesis.
- ”To my knowledge, increasing N_eff to 4 can lift the Hubble constant from CMB in LCDM to a larger value (so smaller H0 tension). However it is stated in line 329 that adding one light sterile neutrino makes the tension worse. Can the author further comment on what is difference here?”
There are several studies devoted to attempts to solve the Hubble tension via the additional relativistic degrees of freedom. We have added comments and discussion on the difference of our result to the corresponding section. In short, the main difference between the results obtained in our work and in other articles is that the nature of Neff is different. In our case, when adding a light sterile neutrino, Neff is equal to 4, since an additional relativistic particle physically exists. In other reviews, Neff is equal to 4 due to the presence of some additional physical effects (for example, heating of active species due to the decay of heavy sterile species, large lepton asymmetry, etc.). Therefore, the resulting effects of these Neff are different.
- “It seems Tab.1 reports 100*Omega_i rather than Omega_i .”
We have changed the caption of the Table accordingly.
- “In the footnote below Tab.1, SH0ES measurement of H0 is not entirely independent of LCDM. In order to extract H0 from the observed luminosity distance d_L(z), a model of d_L(z) needs to be assumed.”
We agree, that some model of d_L(z) should be assumed. However, for small redshifts (e.g. SH0ES collaboration uses objects with the mean redshift <z>~0.055) d_L(z) relation can be expanded into series with leading term being independent of all cosmological parameters but H0: d_L(z) = cz / H0. This is addressed in SH0ES paper (Riess, A. et al. ApJ 934 L7 2022) in sections 2 and 5.2 (direct quote: “This empirical approach is fully independent of the CMB, so an independent comparison of H0 to the CMB with ΛCDM is appropriate”) and this is the reason why we state that this measurement is independent of LCDM.
Reviewer 2 Report
Comments and Suggestions for Authors
The paper is a short review on the role of neutrinos in cosmology. After a brief review on neutrino properties and their role in the Standard Model the authors describe the consequences on cosmological observables (in particular, the primordial element abundances). The authors point out that a the measure of the non thermal background of neutrinos coming from BBN would give valuable direct information on the early stages of the Universe.
Consequences of introducing new sterile states are discussed. In particular it is shown that the simultaneous presence of a sterile neutrinos and a neutrino-antineutrino asymmetry can solve the H0 tension without spoiling the concordance between the observed and the predicted 4He abundance.
In my opinion, the paper, although is far from a complete review on the topic, is a good introduction to a non expert audience.
Author Response
The authors thank the reviewer for careful reading of the paper. Having paid attention to the comment "far from a complete review on the topic" we have slightly expanded the review and added to the list of references.
Round 2
Reviewer 1 Report
Comments and Suggestions for Authors
The authors have successfully addressed all concerns in the previous report and revised the manuscript accordingly. I now recommend the manuscript for publication.